# Economic Evaluation of Newborn Screening for Severe Combined Immunodeficiency

**DOI:** 10.3390/ijns8030044

**Published:** 2022-07-20

**Authors:** Sophy T. F. Shih, Elena Keller, Veronica Wiley, Melanie Wong, Michelle A. Farrar, Georgina M. Chambers

**Affiliations:** 1Surveillance, Evaluation and Research Program, Kirby Institute, University of New South Wales, Sydney, NSW 2052, Australia; 2National Perinatal Epidemiology and Statistics Unit, Centre for Big Data Research in Health and School of Clinical Medicine, University of New South Wales, Sydney, NSW 2052, Australia; e.keller@unsw.edu.au (E.K.); g.chambers@unsw.edu.au (G.M.C.); 3NSW Newborn Screening Programme, Children’s Hospital Westmead, Westmead, NSW 2145, Australia; veronica.wiley@health.nsw.gov.au; 4Faculty of Medicine and Health, University of Sydney, Sydney, NSW 2006, Australia; 5Department of Allergy and Immunology, Children’s Hospital at Westmead, Westmead, NSW 2145, Australia; melanie.wong@health.nsw.gov.au; 6Department of Neurology, Sydney Children’s Hospital, Randwick, Sydney, NSW 2031, Australia; m.farrar@unsw.edu.au; 7Discipline of Paediatrics, School of Clinical Medicine, UNSW Sydney, Sydney, NSW 2052, Australia

**Keywords:** SCID, newborn screening, cost-effectiveness, Markov model

## Abstract

Evidence on the cost-effectiveness of newborn screening (NBS) for severe combined immunodeficiency (SCID) in the Australian policy context is lacking. In this study, a pilot population-based screening program in Australia was used to model the cost-effectiveness of NBS for SCID from the government perspective. Markov cohort simulations were nested within a decision analytic model to compare the costs and quality-adjusted life-years (QALYs) over a time horizon of 5 and 60 years for two strategies: (1) NBS for SCID and treat with early hematopoietic stem cell transplantation (HSCT); (2) no NBS for SCID and treat with late HSCT. Incremental costs were compared to incremental QALYs to calculate the incremental cost-effectiveness ratios (ICER). Sensitivity analyses were performed to assess the model uncertainty and identify key parameters impacting on the ICER. In the long-term over 60 years, universal NBS for SCID would gain 10 QALYs at a cost of US $0.3 million, resulting in an ICER of US$33,600/QALY. Probabilistic sensitivity analysis showed that more than half of the simulated ICERs were considered cost-effective against the common willingness-to-pay threshold of A$50,000/QALY (US$35,000/QALY). In the Australian context, screening for SCID should be introduced into the current NBS program from both clinical and economic perspectives.

## 1. Introduction

Severe combined immunodeficiency (SCID) is a primary immunodeficiency (PID) characterized by a severe deficiency of T-cells. It is a rare inherited disease affecting between 1 in 50,000 and 1 in 100,000 newborns. Two-thirds of cases are X-linked, meaning that it predominantly occurs in males [1]. While affected babies typically show no signs of the disease at birth, without treatment they often do not survive past the first two years of life due to recurrent infections. Viral and fungal infections tend to occur from the age of two months, and bacterial infections emerge from the age of four to six months when the trans-placentally acquired maternal antibodies disappear. Respiratory tract and gastrointestinal infections are most common, leading to malnutrition and impaired growth [2]. Approximately 50% of children with SCID present with a life-threatening respiratory tract infection secondary to Pneumocystis jirovecii (PJP), typically between 2 to 6 months of age [3]. A mortality rate of 35% has been reported in immunocompromised children with PJP infection [4].

Most infants with SCID are diagnosed within the first year of life and require a hematopoietic stem cell transplant (HSCT) to survive. Early diagnosis by newborn screening for SCID allows for HSCT to be undertaken before infections cause any complications that may be life threatening. In the absence of family history of SCID and without a population-based universal newborn screening (NBS) program for SCID, babies with SCID often present with severe infections between three and six months; if untreated, infants with SCID succumb early in life from severe and recurrent infections., Only around 20% of affected babies with a positive family history for SCID are likely to be identified in the absence of population-based NBS [5]. Early diagnosis prior to any infections is crucial to achieve the best long-term outcomes in affected patients in terms of reducing the burden of recurrent infections and providing the opportunity to treat the underlying cause.

The most common treatment is hematopoietic stem cell transplantation (HSCT) from a related or unrelated registry-derived donor. The survival rate after HSCT is highest in infection-free infants treated within the first 3.5 months of life, with 94% of patients alive after 2 years. In older infants, survival is mainly determined by the absence of infections with 90% of infection-free infants alive two years after HSCT compared to 82% of those with resolved infections and 50% of those with active infections during transplantation [6]. Therefore, the main predictors of long-term survival of SCID patients are early treatment with HSCT and good patient health at the time of transplantation indicated by the absence of active infections.

Given the lack of symptoms at birth together with the severity of SCID and the availability of effective treatment if initiated early, the disease is an ideal candidate for NBS [7]. All traditional criteria from Wilson and Junger’s WHO Principles of early detection have been satisfied [8]. NBS for SCID will enable the provision of a definitive diagnosis, avoiding the diagnostic odyssey that families commonly encounter. Importantly the benefits of early diagnosis and initiation of treatment on the final outcome are now recognized, with the most beneficial response to treatment to date seen in patients treated before the onset of symptoms. Considering the technology advancement since the Wilson and Jungner screening criteria published in 1968, SCID is qualified in many aspects of emerging screening criteria in the era of genetic screening [9]. Scientific evidence of screening program effectiveness and cost-effectiveness has been demonstrated in many countries [1,10,11,12,13,14,15]. The available testing by dried blood spot (DBS) NBS using a T-cell receptor excision circle (TREC) assay to determine the number of naïve T-cells has been shown to be effective in diagnosing SCID. The TREC assay is a high-throughput test using DNA based technology. It is extremely sensitive and specific, meaning that false negative and false positive results are unlikely [7] depending on the action limits used. Most importantly, the evaluation has been planned from the outset with an assessment of benefits and harms (including the economic benefits presented in this paper) and an understanding of parents/carers perspectives in the decision making being undertaken through the pilot program evaluation projects. The NBS evaluation study was proposed to assess those newborns identified and then confirmed to have either SMA or PID, through the NBS pathway; feasibility of the screening procedure, temporal course of the various steps of screening and the parental attitude towards screening for SMA and PID were assessed.

Cost-effectiveness analyses of NBS for SCID have been conducted in other countries such as the United States, United Kingdom, New Zealand and Netherlands [10,11,12,14,15,16,17,18]. However, cost-effectiveness of NBS for SCID in the Australian context is lacking. Therefore, to inform future policy development, our objectives were to evaluate the cost-effectiveness of introducing NBS for SCID by modelling the long-term costs and health benefits based on a NBS pilot program for PID in the Australian States of New South Wales (NSW) and Australian Capital Territory (ACT) [19].

## 2. Materials and Methods

Since August 2018, a statewide population-based combined spinal muscular atrophy (SMA) and primary immunodeficiency (PID) newborn screening pilot program has been conducted in the Australian state of NSW and ACT to detect infants at risk of SMA or PID. The latter specifically screens for Severe Combined immunodeficiency (SCID) and B cell deficiency. The SMA/PID NBS has two parts (a) the screening in the NBS laboratory, and, where indicated, (b) the further diagnostic assessment. Up until July 2020, 202,388 newborns in these two states were screened for both SMA and PID in a combined testing panel [20]. These two states approximate one third of all births in Australia where NBS uptake is extremely high with over 99% of live births screened. The second part of diagnosis was performed by clinicians with consultation with NBS and dedicated SMA/PID centers. The clinician will send a repeat dried blood spot to NBS for testing and a separate blood sample to a different testing laboratory.

We conducted the economic evaluation from the Australian government as the payer perspective to compare the costs and quality-adjusted life-years (QALYs) of early diagnosis of SCID by NBS with early HSCT to those of late diagnosis of SCID by clinical symptoms without NBS treated with late HSCT. We took a stepwise approach to our cost-effectiveness analyses. Firstly, we developed Markov cohort simulations reflecting the SCID disease states for (1) pre-symptomatic detection of SCID and early treatment with HSCT and (2) clinical detection of SCID and late HSCT treatment after 3.5 months of age or later, and undertook cost-effectiveness analysis comparing these scenarios. We then embedded these Markov cohort simulations within a decision analytic model to evaluate the cost-effectiveness of introducing NBS for SCID with early treatment by HSCT compared to no NBS program for SCID. All of the analyses were undertaken using the TreeAge Pro 2020 software (TreeAge Software, Williamstown, MA, USA).

The decision analytic model was informed by screening results from the NSW/ACT NBS pilot program in Australia, and the Markov models were based on published literature. Published cost-effectiveness analyses of NBS for SCID are summarized in Table 1. The structure of the Markov models for SCID after early (with NBS) and late diagnosis (without NBS) were based on Van der Ploeg, Blom [17] and Chan, Davis [16]. Six health states that reflect disease progression were defined: pre-symptomatic SCID, HSCT, SCID ‘Well’, SCID ‘Moderate’, SCID ‘Poor’, and deceased. (Figure 1)

We defined the length of each Markov cycle as three months, with half cycle corrections, to capture the distinctions in disease progression and the natural disease history experienced by early and late diagnosis. The disease progression was modelled over a time horizon of 5 and 60 years to consider both short- and long-term costs and health outcomes. Generally, after each Markov cycle, patients could either remain in the current health state or transition to another health state. Model parameters, values and their sources of valuation are presented in Table 2. In our models for early and late diagnosed SCID, all SCID-positive newborns started the Markov cohort simulation in the pre-symptomatic state (Health State in Figure 1).

### 2.1. Intervention Markov Model (NBS and Early HSCT Treatment)

In the NBS intervention cohort, modelling assumed all infants diagnosed with SCID transition to treatment after the first cycle, defined as three months duration, to capture the benefits of early treatment with HSCT. Transition probabilities to the SCID Well, Moderate and Poor states after treatment were based on the published literature [17], using the probability to survive until treatment [1,11,22] as well as 5-year survival [23] from the literature with mortality of the general population applied thereafter. Treatment effectiveness assumed that 80% of infants progress to the SCID Well state, 15% to the SCID Moderate state and 5% to the SCID Poor state after treatment [17]. In this context, SCID Well required an immunologist visit and laboratory tests once every five years. Similarly, SCID Moderate required an immunologist visit and laboratory tests once every three years, with 68% of patients in this health state requiring immunoglobulin therapy. Finally, SCID Poor required an annual immunologist visit and laboratory test with 36% of subjects in this health state requiring immunoglobulin therapy. We validated the model by assessing survival outputs of early SCID and late SCID from our Markov models against those reported in the literature.

### 2.2. Comparator Markov Model (Late HSCT without NBS)

In the cohort without NBS for SCID, we assumed that all infants first developed infections (i.e., transition to SCID Poor), except for 20% of newborns with a family history that was identified at birth, before being diagnosed and treated with HSCT after two cycles. For all of the newborns, we assumed that they start out in the pre-symptomatic state (i.e., cycle 1 for all babies is in pre-symptomatic state). Early treatment in our model after NBS took place in the second cycle; therefore, late treatment occurred after the second cycle. Specifically, for the comparator of late HSCT without NBS, from the second cycle, the infants (aged 3 months+) entered into SCID poor before they received HSCT and survived from infections. We applied the probability to survive until treatment [11] as well as the 5-year survival [23] based on the published literature, and mortality of the general population thereafter was applied. Transition probabilities for the SCID Well, Moderate and Poor states after treatment (50%, 30% and 20%, respectively) were based the Van der Ploeg, Blom [17]. The health utilities and disease progression of Well, Moderate and Poor health states were defined the same as in the intervention Markov model and model validation was equivalent.

### 2.3. Quality of Life

Each health state in the Markov model was assigned a health-related quality of life (HRQoL) utility value to generate QALYs. Utility values for the SCID Well, Moderate and Poor health states were derived from the literature, 0.95, 0.75 and 0.5, respectively [17]. For pre-symptomatic SCID we assumed the same utility value as in the SCID Well state (i.e., 0.95) and during HSCT treatment we assumed a utility value equal to the SCID Poor state (i.e., 0.5). The QoL utility value in the death state was 0.

### 2.4. Costs

Costs considered in the models included screening, investigation (including true and false positives), diagnosis, HSCT treatment, and direct medical care. The screening costs were provided from the NSW/ACT pilot NBS program [20]. All other costs were based on Van der Ploeg, Blom [17], adjusted for the three-month cycle lengths. All costs were collected in Australian dollars and converted to 2018 US$ based on Purchasing Power Parities (PPP) published by OECD [24].

Diagnostic testing either after a positive screening test or following clinical suspicion of SCID, included a pediatrician visit, flow cytometry including a clinic visit, repeat flow cytometry for two-thirds of infants screening positive (only considered in intervention Markov model) and genetic tests. Costs for HSCT treatment were higher after late diagnosis due to poorer outcomes of patients (Table 2). Infants dying prior to HSCT were assumed to develop severe infections requiring expensive medical care totaling US$178,923 (A$259,617). Based on the definitions of the SCID Well, Moderate and Poor states, we assumed annual costs associated with SCID care of US$34 (A$50), US$24,052 (A$34,900), and US$12,873 (A$18,679), respectively. Given that subjects in the SCID Moderate and Poor health states were likely to develop severe lung disease, a one-off additional end-of-life cost of US$41,841 (A$60,712) was included in the model.

This study has been approved by The Sydney Children’s Hospitals Network Human Research Ethics Committee (reference number LNR/18/SCHN/307). Parents of the children participating in the NBS pilot evaluation study provided written consents.

### 2.5. Cost-Effectiveness Analysis

The costs and QALYs were estimated for one infant diagnosed with SCID for a time horizon of 5 and 60 years. Incremental costs and incremental QALYs were estimated as the differences between NBS with early HSCT versus no NBS with late HSCT after the development of symptoms. Incremental costs and incremental QALYs were compared to calculate the incremental cost-effectiveness ratios (ICER) and reported as the incremental cost per QALY.

### 2.6. Sensitivity Analysis

A one-way sensitivity analysis was performed to identify the parameters with a significant impact on the resulting ICER. A tornado diagram was created to visualize the effect of these parameters on the ICER. In addition, probabilistic sensitivity analysis (PSA) using Monte Carlo simulation with 1000 iterations was conducted using relevant model parameters with distributions based on published literature (Table 2). Based on the results of the PSA, cost-effectiveness acceptability curves and cost-effectiveness planes with 95% confidence intervals (CI) were generated.

## 3. Results

### 3.1. Cost-Effectiveness of SCID Treatment Strategies Using Markov Simulation

Total and incremental costs and QALYs for one infant diagnosed with SCID and treated with either pre-symptomatic or late initiation HSCT and the resulting ICERs comparing NBS to no NBS from the government perspective over 5-year and 60-year time horizons are presented in Table 3.

From a government perspective over a 5-year time horizon, pre-symptomatic treatment of a SCID patient with early HSCT would result in 3.50 QALYs at a total cost of US$135,624 (discounted at 3%). Late treatment with HSCT for one symptomatic SCID patient would only result in 1.97 QALYs at a total cost of US$258,133. Therefore, early treatment with HSCT dominated late treatment with HSCT, that is, early HSCT resulted in 1.53 more QALYs per patient diagnosed with SCID with cost savings of US$122,509 per child with SCID over the 5-year time projection. Over a 60-year time horizon, early treatment with HSCT also dominated late treatment, with 6.14 more QALYs gained per patient diagnosed with SCID, at savings of US$136,914 per patient.

### 3.2. Cost-Effectiveness of NBS for SCID Including Treatment Strategies

Over a 5-year time horizon, screening every newborn in the population and treating diagnosed SCID patients with early HSCT would cost US$8.19 and result in 0.00007 QALYs per infant in the population (US$0.8 million and 7 QALYs in 100,000 infants). Compared to clinically diagnosed SCID treated with late HSCT without NBS, NBS for SCID would achieve 2 QALYs gained in 100,000 infants at a cost of US$0.35 million over 5 years from the government perspective, resulting in an ICER of US$144,000 per QALY gained for a universal NBS program for SCID.

Over the 60-year time horizon, screening every newborn in the population and treating diagnosed SCID with early HSCT would result in 0.0001 more QALYs gained, compared to no NBS, at a marginal additional cost of US$3.28 (10 QALYs gained at additional cost of US$0.3 million in 100,000 infants), resulting in an ICER of US $33,600 per QALY gained in a universal NBS program for SCID.

### 3.3. Sensitivity Analysis for the SCID NBS Cost-Effectiveness Analysis

Results of the PSA for the SCID NBS cost-effectiveness using 1000 iterations of Monte Carlo simulations of costs and effectiveness are presented in Figure 2. The associated cost-effectiveness acceptability curve is presented in Figure 3. Simultaneously varying key parameters of screening costs and probabilities identified by a one-way sensitivity analysis for the 60-year time horizon produced ICERs with a 95% CI of US$18,000/QALY to US$59,000/QALY. Fifty-five percent of simulations were under the common willingness-to-pay threshold of A$50,000/QALY (US$35,000/QALY) in the Australian decision-making context (approximately as illustrated in the acceptability curve in Figure 3 and the incremental cost-effectiveness plane in Figure 4).

The results of the one-way sensitivity analyses are represented in the tornado diagram in Figure 5 demonstrating the relative impact of selective parameters on the ICERs. The most significant impact on the ICER resulted from the SCID incidence and discount rate, followed by percentage of early diagnosis by family history without universal NBS and the specificity of the SCID screening test.

## 4. Discussion

The cost-effectiveness analysis results of treating SCID pre-symptomatically with HSCT compared to treating an infant who has become symptomatic (3.5 months or older) strongly suggest that early treatment is cost-saving from a government perspective, compared to thresholds for rare and ultra-rare condition such as SCID (US$500,000/QALY proposed by Institute for Clinical and Economic Review [27]. That is, more health-related QALYs are gained at a lower cost under both a 5-year and 60-year time projection. Based on our Markov modelling cost-effectiveness results, early diagnosis of SCID and treatment with HSCT before symptom onset is warranted and justified for universal screening.

By incorporating NBS into the analysis, the ICER results suggest that a population-based NBS for SCID is cost-effective, using a traditional willingness-to-pay threshold in the Australian decision-making context of A$50,000 (approximately US$35,000/QALY) over a 60-year timeframe. However, over a 5-year time horizon the ICER of US $144,000 per QALY gained exceeds the traditional willingness-to-pay thresholds. This is because the long-term survival difference between early and late initiation of HSCT has not been fully captured by the 5-year analysis. However, despite this relatively high ICER over a short time horizon, it would likely still be considered cost-effective for a novel therapy under ultra-rare disease which typically adopt higher willingness-to-pay thresholds ranging from US$50,000 to US$500,000 per QALY [28,29,30,31].

It is difficult to directly compare the cost-effectiveness analyses presented here with other studies using different definitions of costs and outcomes; however, our study results over the long-term are comparable. For instance, Bessey et al. [12] reported an ICER of US$26,409 per QALY gained for SCID NBS in the UK and also concluded that SCID NBS is cost-effective, while Chan, Davis [16] reported an ICER of US$27,907/QALY in 2013 US value based on US data. In contrast, Van der Ploeg, Blom [17] reported a slightly higher ICER of €33,400 per QALY gained (US$39,433/QALY) which is still considered cost-effective for rare diseases. Using an alternate approach, Ding et al. reported their base-case model suggesting an ICER of US$35,311 per life-year saved, and a benefit-cost ratio of either 5.31 or 2.71 using values of US$4.2 and US$9.0 million per death averted, respectively, which also suggests good value for money of introducing SCID screening into NBS programs [11].

We evaluated and validated our SCID Markov models with published studies to test their validity. The survival outputs from our Markov model closely matched survival reported in observational studies and used in cost-effectiveness analysis in US and Europe [11,17]. However, our study limitations should be considered when interpreting the results. Firstly, for evaluation purposes, our economic analysis focuses on SCID rather than the spectrum of PID. We did not include the costs and benefits of these early identified non-SCID PID cases in our models although the screening pathway of the state-wide NBS identifies non-SCID PID. Approximately one third of non-SCID PID are healthy at birth and therefore would also benefit from early identification and commencement of prophylactic therapy to prevent infection and its impacts [1]. On the other hand, additional costs and harm of detection of mild variants or non-symptomatic variants may arise due to unnecessary investigation and anxiety. However, one UK cost-effectiveness analysis study on NBS for SCID, taking into account investigation and treatment costs of non-SCID PID, concluded that NBS is cost-effective [12]. If costs and health benefits of early intervention for non-SCID PID had been included in our evaluation, NBS including non-SCID PID would likely remain cost-effective. Lastly, there is a lack of Australian specific data on SCID and therefore the latest available literature was applied in our model. We believe there is little variations in terms of disease progression for SCID. However, the factors determining the cost-effectiveness, such as the effectiveness of screening program (e.g., detection rate) and cost of screening test, were Australian specific based on our pilot program.

Disease incidence is an influential factor as indicated by the one-way sensitivity analysis. The higher the incidence of a disease, the more cost-effective a NBS program is. The proportion of SCID identified early without screening based on family history also impacts on the ICER. Costs of the screening test in a universal screening program are crucial for determining its cost-effectiveness, in particular for rare or ultra-rare diseases. However, the screening cost for SCID in our analysis does not appear to be a determinant variable. A systematic review indicated the average cost of NBS for SCID using TREC varied between US$3–US$6 [32]. Adding the TREC assay for SCID alone to the current NBS program in Australia would cost an additional A$7 (US$4.8) per infant screened, which is comparable to costs reported globally. Multiplex testing is anticipated to reduce the unit cost. SCID was the first condition in NBS that used a molecular based test targeting non-replicating DNA fragments in naïve T-cells for the primary screen. Studies have shown the reliability and analytical accuracy of multiplex PCR assays for spinal muscular atrophy (SMA) and SCID in a single analytical process [33,34]. Including SMA with existing SCID screening may be as little as an additional US$1 per specimen [35]. This approach has already been implemented in some global NBS programs [36]; however, cost-effectiveness analyses of a multiplex assay in a NBS program for multiple diseases is required to demonstrate its efficiency and cost-effectiveness.

Although the cost of a screening test is crucial, the consequent diagnosis procedures are also significant to determine the efficiency and cost-effectiveness of universal screening for SCID. Based on the real-life data from a prospective NBS implementation study in the Netherlands, van den Akker-van Marle et al. suggest that strategies with a lower number of referrals, e.g., by distinguishing between urgent and less urgent referrals, are favorable from an economic point of view [14]. The sensitivity analysis of our modelled economic evaluation also indicates the screening test specificity, defining the rate of true negatives detected by NBS, is an important factor influencing the ICERs. Premature infants have low TRECs, and false negatives may also occur with transfusion prior to sample collection. In T- and B-cells, re-arrangement of the T-cell receptor and immunoglobulin loci gives rise to circular DNA fragments (TRECs) and Kappa-depleting Recombination Excision Circles (KREC), respectively. Quantitative PCR amplification can therefore be used to establish the relative copy number of these species in NBS allowing newborns with T- and/or B-cell lymphopenia to be distinguished from healthy infants. In our pilot NBS program for PID, most false positives were due to KREC assay to detect B-cell lymphopenia rather than TREC for T-cell lymphopenia. Our pilot program has modified the follow-up action pathway and hence resulted in a significant reduction in false positive notifications of potential PID.

In conclusion, our analysis suggests that NBS for SCID with early HSCT treatment would be considered cost-effective from the government perspective. A universal NBS program for SCID is supported not only by its clinical values but also by its economic benefits.

## Figures and Tables

**Figure 1 IJNS-08-00044-f001:**
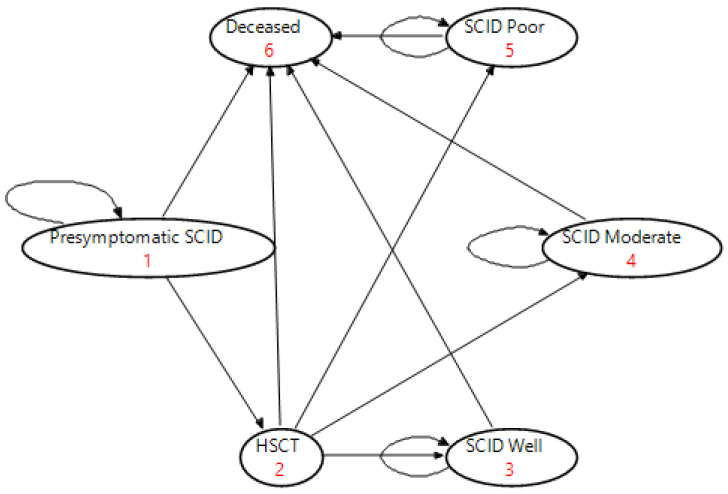
Health state transition diagram for SCID Markov model.

**Figure 2 IJNS-08-00044-f002:**
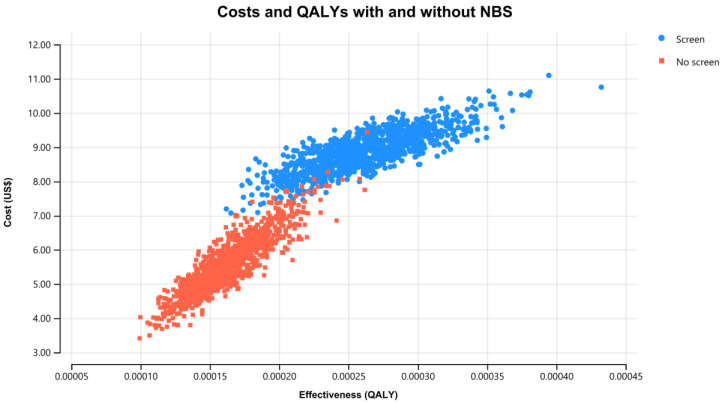
Costs and QALYs for SCID NBS and no SCID NBS, government perspective, 60 years, discounted US$ (d = 3% p.a.).

**Figure 3 IJNS-08-00044-f003:**
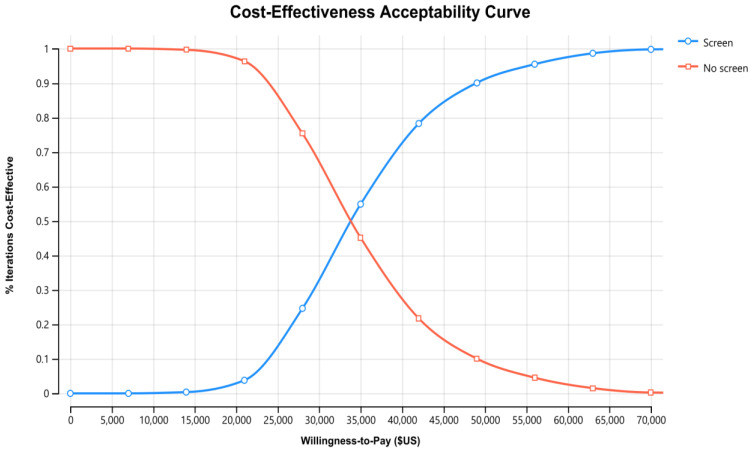
Cost-effectiveness acceptability curve for SCID NBS, government perspective, 60 years, discounted US$ (d = 3% p.a.). Note: At the willingness-to-pay threshold of US$35,000 per QALY, the percentage of simulated ICERs of screen (blue line) is about 55%, whereas at the willingness-to-pay threshold of US$50,000 per QALY, the screen intervention (blue line) has 90% chance and above to be cost-effective.

**Figure 4 IJNS-08-00044-f004:**
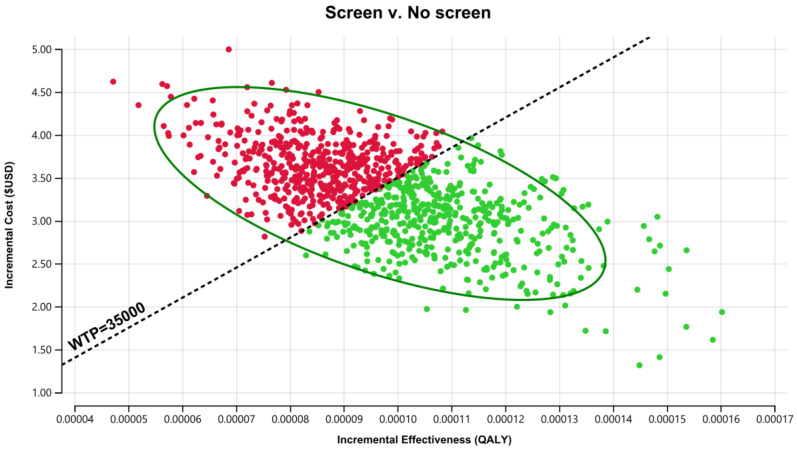
Cost-effectiveness plane for SCID NBS vs. no NBS, government perspective over 60 years, discounted US$ (d = 3% p.a.). (Note: Green dots represent iterations considered cost-effective (ICER less than the willingness-to-pay (WTP) threshold US$35,000/QALY); while red dots represent iterations considered not to be cost-effective (ICER greater than the WTP threshold).

**Figure 5 IJNS-08-00044-f005:**
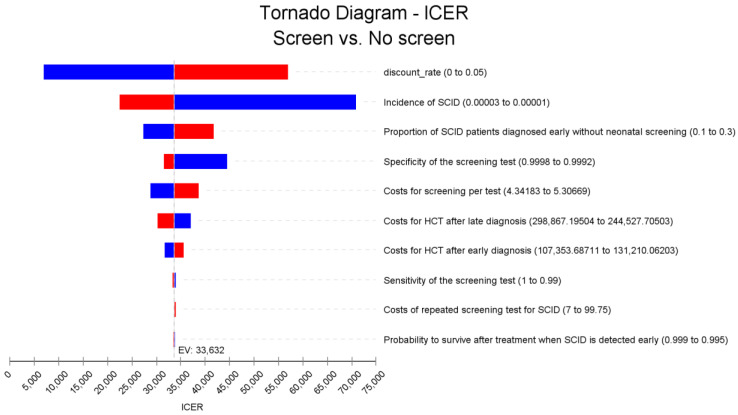
One-way sensitivity analysis for SCID NBS cost-effectiveness analysis, government perspective. Note: Red bars indicate an increase in the parameter value from the base case value (Expected Value, EV line) and the blue bars show otherwise.

**Table 1 IJNS-08-00044-t001:** Published cost-effectiveness analyses of newborn screening (NBS) for severe combined immunodeficiency (SCID).

Source	Country	Results
Chilcott, Bessey [21]	United Kingdom	Detection of 17 affected newborns annually; total gain of 184 discounted QALYs; annual costs of SCID NBS approximately £3.0 million; ICER of £17,600 per QALY gained
Bessey, Chilcott [12]	United Kingdom	Screening for SCID was estimated to result in an incremental cost-effectiveness ratio (ICER) of £18,222 with a reduction in SCID mortality from 8.1 (5–12) to 1.7 (0.6–4.0) cases per year of screening.
Chan, Davis [16]	USA	Over a 70-year time horizon, the average cost per infant was US$8.89 without screening and US$14.33 with universal screening, ICER US$27,907/QALY. The model predicted that universal screening in the U.S. would cost approximately US$22.4 million/year with a gain of 880 life years and 802 QALYs.
Ding, Thompson [11]	USA (Washington State)	Additional 1.19 newborn infants with SCID detected preclinically through screening and 0.40 deaths averted in an annual birth cohort of 86,600 infants. Base-case model suggests an ICER of US$35,311 per life-year saved, and a benefit-cost ratio of either 5.31 or 2.71.
Mcghee, Stiehm [18]	USA	A nationwide screening program would cost an additional US$23.9 million per year for screening costs but would result in 760 years of life saved per year of screening. The cost to detect 1 case of SCID would be US$485,000.
Health Partners Consulting Group [10]	New Zealand	Adding newborn screening for SCID to the Newborn Metabolic Screening Program (NMSP) may result in saving of 10.0 life years at a cost of NZ30,000 per life-year.
Van der Ploeg, Blom [17]	Netherlands	The number of deaths due to SCID per 100,000 children was assessed to decrease from 0.57 to 0.23 and 11.7 quality adjusted life-years (QALYs) gained was expected. Total healthcare costs were €390,800 higher in a situation with screening compared to a situation without screening, resulting in a cost-utility ratio of €33,400 per QALY gained.
Van den Akker-Van Marle [14]	Netherlands	Cost-effectiveness ratios varied from € 41,300 per QALY for the screening strategy with T-cell receptor excision circle (TREC) ≤ 6 copies/punch to € 44,100 for the screening strategy with a cut-off value of TREC ≤ 10 copies/punch

**Table 2 IJNS-08-00044-t002:** Model parameter and expected values with ranges for Markov model ($ 2018 value).

Parameters	Expected Values	Distribution	Low	High	Source
Cost Parameters	US$ (A$)				
Screening test cost	$4.82 ($7)	Gamma	−10%	+10%	NBS Pilot
Repeat screening test cost	$6.89 ($10)	Gamma	7	25	NBS Pilot
Confirmatory diagnostic testing cost (after positive screening test)	$2119 ($3074)				[17]
Diagnostic testing cost (without NBS)	$3446 ($5000)				[17]
Pre-symptomatic cost	$0 ($0)				Assumption
HSCT cost (early diagnosis)	$119,282 ($173,078)	Gamma	−10%	+10%	[17]
HSCT cost (late diagnosis)	$271,697 ($394,233)	Gamma	−10%	+10%	[17]
Treatment cost for SCID patient dying prior to HSCT	$178,923 ($259,617)				[17]
SCID Well treatment cost (per year)	$34 ($50)				[17]
SCID Moderate treatment cost (per year)	$24,052 ($34,900)				[17]
SCID Poor treatment cost (per year)	$12,873 ($18,679)				[17]
End-of-life costs (SCID Moderate & SCID Poor)	$41,841 ($60,712)				[17]
SCID Well productivity cost	$0				Assumption
SCID Moderate productivity cost	$1394 ($2023)				[17]
SCID Poor productivity cost	$0 ($0)				Assumption
Netherlands, 2016 PPP Euros/US$	0.796				[24]
Australia, 2016 PPP A$/US$	1.45				[24]
CPI inflation rate 2016 to 2019 A$	0.0557				[25]
Discount rate	0.03		-	0.05	
**Outcome Parameters**					
SCID incidence	0.00002	Beta	0.000012	0.000025	[1,26]
False Negative % in Screen (1-sensitivity)	0.005	Beta	0	0.01	[11]
False Positive % in Screen (1-specificity)	0.0003	Beta	0.0002	0.0008	[11]
% of patients early diagnosed without NBS	0.2	Beta	0.1	0.3	[16]
Probability to survive until treatment (early diagnosis)	0.9423				[1]
Probability to survive until treatment (late diagnosis)	0.78				[11]
5-year survival (early diagnosis)	0.94	Beta	0.91	0.98	[23]
5-year survival (late diagnosis)	0.82	Beta	0.7	0.9	[23]
SCID Well after HSCT (early diagnosis; surviving subjects)	0.8				[17]
SCID Moderate after HSCT (early diagnosis; surviving subjects)	0.15				[17]
SCID Poor after HSCT (early diagnosis; surviving subjects)	0.05				[17]
SCID Well after HSCT (late diagnosis; surviving subjects)	0.5				[17]
SCID Moderate after HSCT (late diagnosis; surviving subjects)	0.3				[17]
SCID Poor after HSCT (late diagnosis; surviving subjects)	0.2				[17]
**Quality of Life Utility Value**					
Utility value for pre-symptomatic SCID	0.95				Assumption
Utility value for SCID Well	0.95				[17]
Utility value for SCID Moderate	0.75				[17]
Utility value for SCID Poor	0.5				[17]
Utility value for HSCT	0.5				Assumption
Utility value for Deceased	0				Assumption

**Table 3 IJNS-08-00044-t003:** Cost-effectiveness analysis of SCID treatment strategies and NBS for SCID, over 5 and 60 years from the government perspective, discounted 3% p.a. (US$ 2018).

SCID Treatment Strategies					
Strategy	Cost	Incremental Cost	QALY	Incremental QALY	ICER
5 years	(95% CI)	(95% CI)	(95% CI)	(95% CI)	(95% CI)
Late HSCT for SCID	$258,133 (247,010, 270,047)	-	1.97078 (1.75538, 2.16004)	-	
Early HSCT for SCID	$135,624 (128,784, 142,120)	−$122,509 (−135,775, −108,190)	3.50035 (3.49305, 3.50572)	1.52957 (1.33976, 1.74600)	Dominant (dominant, dominant)
60 years					
Late HSCT for SCID	$306,090 (292,494, 319,609)	-	6.90880 (6.06863, 7.64696)	-	
Early HSCT for SCID	$169,177 (162,819, 175,859)	−$136,914 (−151,178, −121,509)	13.05174 (13.01996, 13.07145)	6.14293 (5.40464, 6.97026)	Dominant (dominant, dominant)
**NBS for SCID**					
**Strategy**	**Cost**	**Incremental Cost**	**QALY**	**Incremental QALY**	**ICER**
5 years	(95% CI)	(95% CI)	(95% CI)	(95% CI)	(95% CI)
No screen with late HSCT	$4.67 (3.37, 6.30)	-	0.00005 (0.00003, 0.00006)	-	-
Screen with early HSCT	$8.19 (7.18, 9.28)	$3.51 (2.48, 4.40)	0.00007 (0.00005, 0.00009)	0.00002 (0.000017, 0.000034)	$144,487 (79,155, 242,744)
60 years					
No screen with late HSCT	$5.57 (4.02, 7.29)	-	0.00016 (0.00012, 0.00021)	-	-
Screen with early HSCT	$8.86 (7.71, 10.08)	$3.28 (2.26, 4.24)	0.00026 (0.00019, 0.00034)	0.00010 (0.00007, 0.00013)	$33,632 (17,897, 59,441)

NBS: newborn screening; SCID: severe combined immunodeficiency; ICER: incremental cost-effectiveness ratio.

## Data Availability

Not applicable.

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
