# Peer review of "Economic Evaluation of Newborn Screening for Severe Combined Immunodeficiency"

_2409-515X, 2022, doi:10.3390/ijns8030044_

Round 1
Reviewer 1 Report
Please improve the used definitions, explain why NL data are also comparable to the Australian situation and indicate whether the assumption that with no screening there is always late treatment is the case (we think when there is a SCID patient identified, then there will be early treatment for next children in the family also in a situation without screening.

Author Response
Dear Reviewer:
Thank you for the comments that have been addressed in the following response and attached PDF.
Comments and Suggestions for Authors
Please improve the used definitions, explain why are also comparable to the Australian situation and indicate whether the assumption that with no screening there is always late treatment is the case (we think when there is a SCID patient identified, then there will be early treatment for next children in the family also in a situation without screening.
Response:
There is a lack of Australian-specific data on SCID and therefore the latest available literature was applied in our model. We believe there are little variations in terms of disease progression for SCID. The factors determining the cost-effectiveness, such as the effectiveness of the screening program (e.g. detection rate) and cost of screening test, were Australian specific based on our pilot program.
In the comparator model without NBS, we made the following assumptions. So, family history has been included in modeling the comparator without screening.
“In the cohort without NBS for SCID we assumed that all infants first developed infections (i.e. transition to SCID Poor), except for 20% of newborns with a family history that were identified at birth, before being diagnosed and treated with HSCT after two cycles.” (section 2.2 page 4)

Reviewer 2 Report
The manuscript describes a health economic evaluation of newborn screening for SCID in the Australian context, based on a population-based pilot program. It is concluded that screening for SCID is cost-effective and should be included in the current Autstralian NBS program.
General comments:
It would be valuable with some general information on the current NBS program in Australia. How is it organized? How many babies are born? As the current study focusses on marginal costs for adding SCID to an ongoing program, have health economic evaluations been performed for the other diseases that are included?
Have results regarding organization and/or performance of the pilot screening program for SCID been published? Reference is made (15) but this reference is focused on screening for SMA and does not seem to contain details on the SCID pilot.
Minor comments:
P2: The authors state that “only 20% of affected babies with a positive family history for SCID are likely to be identified in the absence of population-based SCID”. On what data was the estimation based, that 20% have a positive family history? Is this known from the pilot study?
P2 It would be relevant to refer to the Wilson and Jungner criteria when arguing that SCID is an ideal candidate for NBS.
P4 The assumed life expectancies seem low, the authors should comment on how these figures were obtained (especially 65 years in the SCID Well group seems low).
Figure 3 and 4 are not very well explained, this could be improved.
Author Response
Dear Reviewer,
Thank you very much for the comments that we have addressed in the attached response document in blue texts and the revised manuscript. Please see the attachment.
Kind Regards,
Sophy Shih
